# Vibrational Property of α-Borophene Determined by Tip-Enhanced Raman Spectroscopy

**DOI:** 10.3390/molecules27030834

**Published:** 2022-01-27

**Authors:** Ping Zhang, Xirui Tian, Shaoxiang Sheng, Chen Ma, Linjie Chen, Baojie Feng, Peng Cheng, Yiqi Zhang, Lan Chen, Jin Zhao, Kehui Wu

**Affiliations:** 1Institute of Physics, Chinese Academy of Sciences, Beijing 100190, China; zhangping@iphy.ac.cn (P.Z.); sxsheng@iphy.ac.cn (S.S.); chma@iphy.ac.cn (C.M.); bjfeng@iphy.ac.cn (B.F.); pcheng@iphy.ac.cn (P.C.); yiqi.zhang@iphy.ac.cn (Y.Z.); lchen@iphy.ac.cn (L.C.); 2School of Physical Sciences, University of Chinese Academy of Sciences, Beijing 100190, China; 3Department of Physics, University of Science and Technology of China, Hefei 230026, China; txr@mail.ustc.edu.cn; 4Department of Chemical Physics, School of Chemistry, University of Science and Technology of China, Hefei 230026, China; ljchen9@mail.ustc.edu.cn; 5CAS Key Laboratory of Strongly-Coupled Quantum Matter Physics, Department of Physics, University of Science and Technology of China, Hefei 230026, China; 6Songshan Lake Materials Laboratory, Dongguan 523808, China

**Keywords:** borophene, vibrational modes, tip-enhanced Raman spectroscopy, scanning tunneling microscopy, density functional theory calculations

## Abstract

We report a Raman characterization of the α borophene polymorph by scanning tunneling microscopy combined with tip-enhanced Raman spectroscopy. A series of Raman peaks were discovered, which can be well related with the phonon modes calculated based on an asymmetric buckled α structure. The unusual enhancement of high-frequency Raman peaks in TERS spectra of α borophene is found and associated with its unique buckling when landed on the Ag(111) surface. Our paper demonstrates the advantages of TERS, namely high spatial resolution and selective enhancement rule, in studying the local vibrational properties of materials in nanoscale.

## 1. Introduction

Borophene is an emerging two-dimensional (2D) material with novel properties, such as structural anisotropy [1], high thermal conductivity [2], metallicity [3,4], possible superconductivity [5] and polymorphism [4,6,7,8]. The unique polymorphism of borophene stems from the tremendous possible arrangements of hexagonal holes (HHs) in a planar triangular lattice. However, among the huge number of 2D borophene polymorphs that have been designed previously [7,9], only a few of them have been realized experimentally [10]. Using molecule beam epitaxial (MBE) to deposit boron on various metal substrates, different borophene polymorphs have been found on Ag(111) [4], Ag(100) [8], Cu(111) [11,12], Ag(110) [13], Au(111) [14], Al(111) [15] and Ir(111) [16,17]. Among them, the most studied phases are the β_12_ and χ_3_ on Ag(111), and both of them can form large-area single phases with appropriate growth conditions [4]. Their structures and properties have been well established with different methods, such as *in-situ* Raman [18], angle-resolved photoemission spectroscopy (ARPES) [19,20] and high-resolution electron energy loss spectroscopy (HREELS) [21].

Among various 2D borophene polymorphs, the α phase with three-fold symmetry and 1/9 HH density is particularly interesting, as it was predicted to be one of the most stable and fundamental borophene structures [9,10]. Previously, Zhong et al. reported the observation of small α-phase domains coexisting with β_12_ and χ_3_ phases in Ag(111) substrate [6]. Recently, Liu et al. reported the observation of bi-layer borophene on Ag(111), which was assigned to two covalently bonded α-layers [22]. In both cases, the α phase only exists in small islands of typically nanometer size, and a complete α-layer is still not available. This poses a great challenge to the understanding of the physical properties of α borophene, as even microscopic characterization techniques usually require samples of micrometer size. Thus, the properties of α borophene remain elusive so far.

In this paper, the vibrational properties of the α borophene were studied by combining scanning tunneling microscopy (STM) with tip-enhanced Raman spectroscopy (TERS). TERS allows one to detect the local vibrational properties with high spatial resolution (<0.5 nm) by the help of the strong localized electric field under the probe tip [23]. We obtained dramatically different Raman spectra from α borophene, as compared with those from other phases in the previous report [18]. DFT calculations reproduce the vibrational modes observed by the Raman spectra well, based on a buckled α-phase model on Ag(111). Our results provide a fundamental data set for further studies of borophene and demonstrate the capability of TERS in the study of local properties of 2D materials

## 2. Method

All STM and TERS measurements were performed at 77 K using a home-made STM-TERS system (located in Institute of Physics, CAS, Beijing, China), the base pressure being 10^−8^ Pa. The single crystalline Ag(111) surface was cleaned by standard cycles of Ar^+^ ion sputtering and annealing at 800 K. Pure boron was evaporated from an e-beam evaporator to the Ag(111) substrate held at 570 K during deposition [4]. The TERS measurement was performed with side illumination and backscattering collection configuration [24]. A 532 nm laser was focused at the tunneling gap using aspheric lens attached to the side of the STM head in the ultrahigh vacuum chamber. The scattered Raman signals were dispersed by 1200 grooves/mm grating and collected by a liquid-nitrogen-cooled charge coupled device (CCD) (SP2300i, Princeton Instrument, Trenton, NJ, USA).

The first-principle calculations were performed within the framework of projector-augmented wave (PAW) method [25], as implemented in Vienna Ab-initio Simulation Package (VASP) [26,27]. The electronic exchange–correlation interaction was described by Perdew–Burke–Ernzerhof (PBE) functional [28], and the van der Waals (vdW) correction was included using DFT-D3 method with Becke-Jonson damping [29]. We set a 500 eV plane-wave cutoff and adopted a 12 × 12 × 1 k-grid to sample the first Brillouin zone of the unit cell. All the atomic structures of borophene were fully relaxed on a two-layer Ag(111) surface until the changes in energy and force between each iteration step were respectively smaller than 10^−8^ eV and 0.001 eV/Å. To avoid the interlayer interaction, a 30Å vacuum interval was set up. With regard to phonon calculations, we employed the frozen-phonon method with 4 × 4 × 1 supercell and 3 × 3 × 1 k-grid. The phonon dispersion was obtained based on the frozen-phonon results with the help of Phonopy [30]. Finally, all the models were shown using VESTA [31].

## 3. Results and Discussion

After the deposition of about 0.8 ML boron atoms on the Ag(111) substrate, the Ag(111) surface was covered mainly with the β_12_ borophene islands, which exhibited parallel Moiré stripes in parallel with the high-symmetry orientations of the Ag(111) substrate. Meanwhile, small domains with hexagonal Moiré patterns are frequently found to coexist with the β_12_ phase, an example of which is shown in Figure 1a. The high resolution STM image in Figure 1b shows the hexagonal structure of this phase, with the lattice constant *a* = 0.52 ± 0.01 nm. This structure is consistent with the previously reported α borophene on Ag(111) [6].

To understand the structure and properties of the α phase, it is worth noting that a completely flat α structure is unstable because of a large negative phonon frequency [9]. Instead, a symmetric and slightly buckled α phase with a vertical distance from the plane of about ±0.16 Å is found to be stable in the freestanding form [9]. The upward buckled boron atom in this model is marked by A, while the downward one is marked by B (Figure 1c). Furthermore, after relaxing the symmetric buckled α phase on the Ag(111) substrate, we found that the vertical distance of two buckled boron atoms from the plane further increased to 0.36 Å and −0.51 Å, exhibiting an asymmetric buckled structure. The electronic band structure of this asymmetric buckled phase is shown in Figure 1d. Its metallic properties are also consistent with previous STS result [6]. The asymmetric buckling is found to be critical for the vibrational properties of α borophene on Ag(111), as will be shown and discussed below.

TERS measurement was performed to obtain the vibrational information from the α borophene. As Figure 2a shows, when the STM tip is far from the surface, the far-field Raman signal is very weak due to the small Raman scattering cross-section of borophene [18]. When the STM probe tip is brought close to the surface of α borophene, a dramatic enhancement of Raman signal is observed, exhibiting a strong increment with the decrease in gap distance. The near-field TERS spectra clearly show a series of characteristic peaks, as illustrated in the background subtracted spectrum (the red curve in Figure 2b). Five strong peaks are found, located at 116.8 cm^−1^, 157.3 cm^−1^, 339.0 cm^−1^, 702.6 cm^−1^ and 920.4 cm^−1^, together with three weak peaks at 406.4 cm^−1^, 446.8 cm^−1^ and 1230.0 cm^−1^. For comparison, the TERS spectrum of α borophene is quite different from that of β_12_ phase (blue curve in Figure 2b), as well as from that of χ_3_ phase reported in our previous study [18]. In particular, we observe significant enhancements of high-frequency peaks over 500 cm^−1^, in contrast with the cases of β_12_ and χ_3_ phases, where only the low-frequency peaks are enhanced [18]. In view of the selective enhancement mechanism of TERS [18,23], only vibration modes that contain out-of-plane components can be enhanced effectively. For completely flat 2D borophene phases, such as β_12_ and χ_3_, their high-frequency vibrational modes contain only in-plane components, and thus cannot be enhanced in TERS [18]. Therefore, we speculate that the obvious enhancement of the high-frequency Raman modes in α borophene implies that these vibrational modes contain out-of-plane components. This perfectly agrees with the fact that the α borophene is significantly buckled on Ag(111), according to DFT calculations.

We also emphasize that our TERS measurement renders Raman spectrum with extremely high spatial resolution. A series of TERS spectra were taken along the yellow line in the STM image shown in Figure 2c, crossing β_12_-α-β_12_ regions as the α domain is surrounded by β_12_ domains. One can see that when the STM tip moves from β_12_ to α phase, the intensity of the characteristic B_3g_^2^ peak from the β_12_ phase drop immediately, accompanied by the appearance of the characteristic 116.8 cm^−1^ peak from α borophene (the right panel of Figure 2c). Therefore, the high spatial resolution of TERS allows us to well separate the Raman signal of α borophene from that of surrounding β_12_ phase, even though the size of the α borophene domain is only a few nanometers.

To account for these TERS peaks, we performed DFT calculations. The phonon spectra of both symmetric buckled α phase and asymmetric buckled α phase were simulated by VASP, respectively. The phonon spectrum of symmetric buckled α phase was found to largely deviate from our experimental TERS spectra. The phonon modes are completely absent in the vicinity of 116.8 cm^−1^, 157.3 cm^−1^, 339.0 cm^−1^, 406.4 cm^−1^, 1230.0 cm^−1^ at the Γ point. In contrast, after relaxing the structure to the asymmetric buckled α phase, its symmetry changes from *D_3d_* to *C_3v_*, causing the change of phonon spectrum. As shown in Figure 3a, the phonon spectrum of asymmetric buckled α phase shows no obvious negative phonon frequencies, indicating a stable structure. Importantly, most peaks in experimental TERS spectra can be assigned to phonon modes at the Γ point, as shown in Figure 3b. A detailed comparison of experimental and simulated peaks is shown in Table 1. The five low-frequency peaks located at 116.8 cm^−1^, 157.3 cm^−1^, 339.0 cm^−1^, 406.4 cm^−1^, 446.8 cm^−1^ peak can be associated with $E^{7}$, $E^{6}$,$\;A_{1}^{4}$,${\;A}_{1}^{3}$,${\;A}_{2}^{3}$ and phonon modes, respectively, within a reasonable error range. The atomic displacements of these peaks, as shown in Figure 3c, are composed of nearly pure out-of-plane vibrational components, which accords with the selective enhancement rule in TERS. For the two high-frequency peaks located at 702.6 cm^−1^ and 920.4 cm^−1^, we can assign them to two phonon modes E^4^ and E^3^. The schematics of atomic displacements show that these two modes are composed of nearly in-plane vibrational components; however, the out-of-plane vibrational components still exist because of the two buckled boron atoms. Therefore, the enhancement of these two peaks in TERS accords with our model.

Finally, the TERS peak located at 1230 cm^−1^ is about 90 cm^−1^ above the highest phonon mode at Γ in the phonon spectrum. To account for this peak, we attribute it to a vibration mode of a supercell. Here, a 4 × 4 × 1 supercell is considered because the periodicity of Moiré pattern of the α phase is about four times that of unit cell. Due to the Brillouin zone folding, the M point of the unit cell will be folded to the Γ point in the new Brillouin zone of the 4 × 4 × 1 supercell. The $E^{1}$ mode at the M point contributes to a supercell vibrational mode with a frequency of 1199.10 cm^−1^ (Figure 3d), which matches our measurement better than any other modes contained in such a supercell. In addition to the frequency consistency, this mode also contains the out-of-plane vibration component. Therefore, it very likely corresponds to the 1230 cm^−1^ TERS peak.

## 4. Conclusions

In conclusion, we determine the characteristic Raman spectrum of α borophene with the help of high spatial resolution of TERS, combining with DFT calculations. All Raman peaks can be well associated with the phonon modes calculated based on an asymmetric buckled α structure. The unusual enhancement of high-frequency Raman peaks in TERS spectra of α borophene is also related to its unique buckling when landed on the Ag(111) surface. Our work provides not only the basic Raman characterization of the highly interesting α borophene, but also demonstrates the high prospect of TERS in studying local vibrational properties of nanoscale structures.

## Figures and Tables

**Figure 1 molecules-27-00834-f001:**
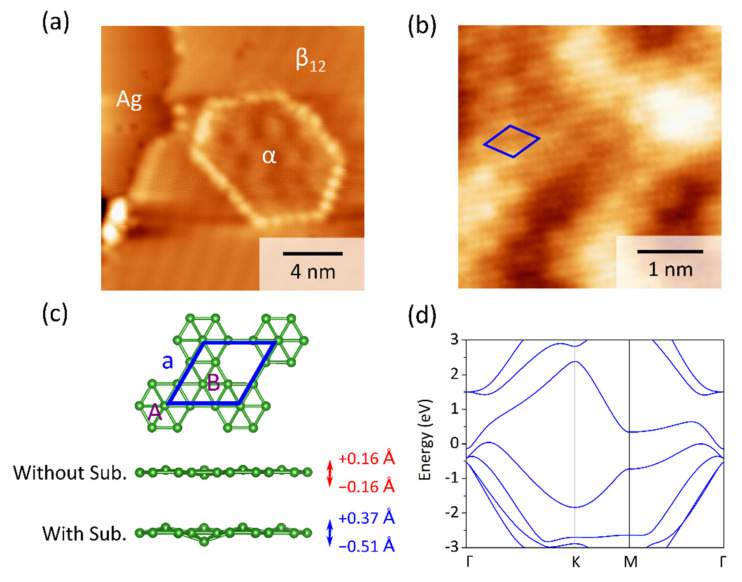
(**a**) STM image (1.3 V, 190 pA) of monolayer borophene sheet grown on Ag(111), showing an α domain between β_12_ islands. (**b**) High-resolution STM image (−0.8 V, 190 pA) of the surface of α borophene; the unit cell is marked by a blue rhombus. (**c**) The model of planar α phase, shown together with the symmetric buckled α phase (relaxed without Ag(111) substrate) and asymmetric buckled α phase (relaxed on Ag(111) substrate). (**d**) The simulated electronic band structure of asymmetric buckled α phase.

**Figure 2 molecules-27-00834-f002:**
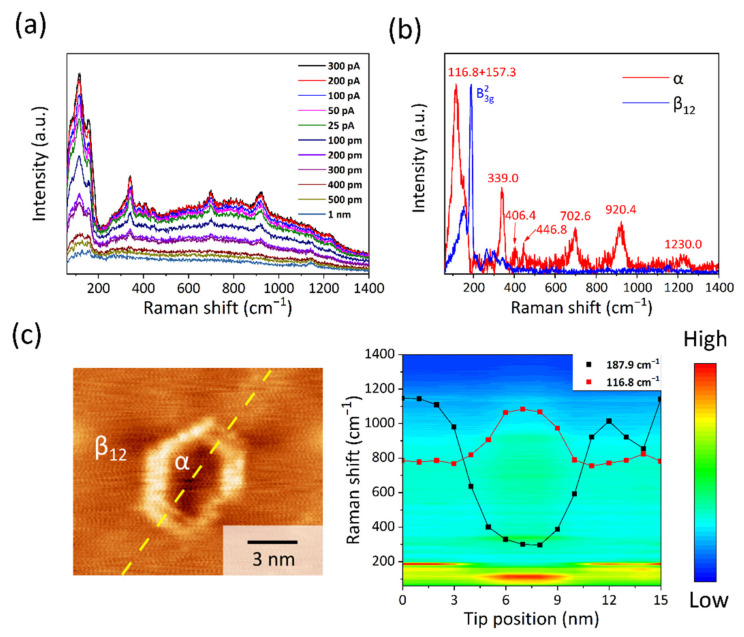
(**a**) Gap distance-dependent TERS spectra of α borophene (10 mW, 0.3 V, the accumulation time for each spectrum is 50 s). The tip-sample distance was controlled by first decreasing the tunneling current from 300 pA to 25 pA, and then the tip was retracted from the surface in 100 pm steps with the feedback loop off. (**b**) Comparison of the TERS spectra of α and β_12_ phases after background subtraction and normalization. (**c**) TERS spectra were taken along the yellow line, crossing the α borophene domain. The Raman intensity map was plotted in the right panel, where two dotted lines are the TERS intensity profiles of the two characteristic peaks of α borophene (red) and β_12_ borophene (black), respectively.

**Figure 3 molecules-27-00834-f003:**
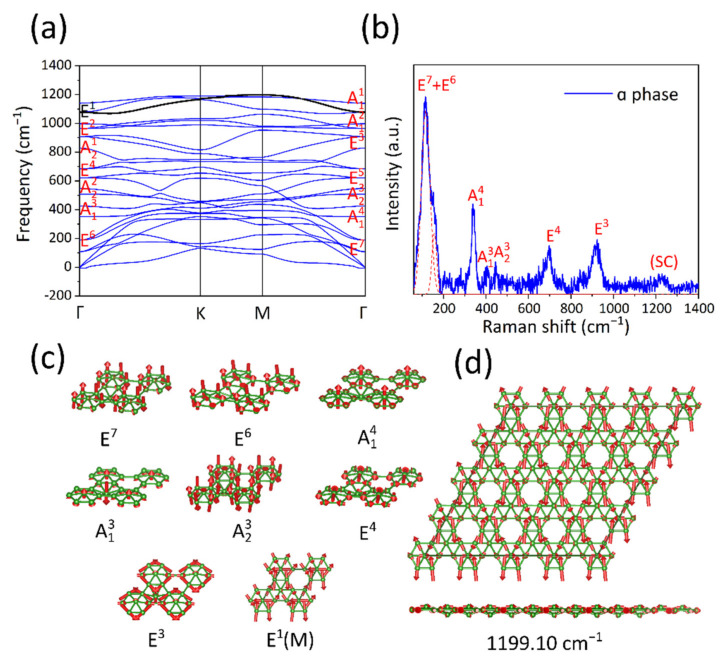
(**a**) The calculated phonon dispersion curves of asymmetric buckled α phase. (**b**) The near-field TERS signal of α phase, E^7^ and E^6^ peaks are fitted by red dashed curve. (**c**) Vibration modes of unit cell associated with TERS peaks. (**d**) The vibration mode of supercell associated with the 1230 cm^−1^ peak in TERS.

**Table 1 molecules-27-00834-t001:** TERS modes of the α phase, as compared with the calculation, cm^−1^.

TERS	Simulation	Modes
116.8	107.85	$E^{7}$
157.3	188.6	$E^{6}$
339.0	352.94	$A_{1}^{4}$
406.4	425.2	$A_{1}^{3}$
446.8	507.83	$A_{2}^{3}$
702.6	684.32	$E^{4}$
920.4	907.16	$E^{3}$
1230.0	1199.10	Vibration of SC

## Data Availability

Data available via personal communication with proper reasons.

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
