# Peer review of "Vibrational Property of α-Borophene Determined by Tip-Enhanced Raman Spectroscopy"

_molecules, 2022, doi:10.3390/molecules27030834_

Round 1

Reviewer 1 Report

The paper is devoted to investigations of the recently founded α polymorph of borophene by scanning tunneling microscopy and tip-enhanced Raman scattering, so the results are extremely interesting for specialists both in material science and in Raman spectroscopy. Reliable data on the structure are obtained, new spectral features are found and interpreted using contemporary ab initio simulations. The extremely high spatial resolution of Raman data and confirmations of the asymmetric buckled structure of the α polymorph should be noted especially. I'm highly impressed. 
Two minor remarks. 
It's not quite clear why the M point at the Brillouin zone was selected to interpret the higher frequency Raman line. According to the symmetry of the buckled structure, any A1 mode is supposed to have some transversal component all over the zone. Focusing on the M point you suppose some special distortion of the structure that activates only this point, but not the whole higher frequency branch. 
Designations of phonons in the table (A4 1, A3 1, A3 2) and at lines 164, 184 should be corrected, to agree with Fig. 3. Check sub- and superscripts.

Author Response

Thank you very much for valuable comments, our responses are included in the attached file.

Reviewer 2 Report

See the File.

Author Response

Thank you for valuable comments. Our responses were included in the attached file to reviewer comment 1
